# Bean Consumption during Childhood Is Associated with Improved Nutritional Outcomes in the First Two Years of Life

**DOI:** 10.3390/nu16081120

**Published:** 2024-04-10

**Authors:** Divya Choudhary, Todd C. Rideout, Amy E. Millen, Xiaozhong Wen

**Affiliations:** 1Division of Behavioral Medicine, Department of Pediatrics, Jacobs School of Medicine and Biomedical Sciences, State University of New York at Buffalo, Buffalo, NY 14214, USA; divyacho@buffalo.edu; 2Department of Exercise and Nutrition Sciences, School of Public Health and Health Professions, State University of New York at Buffalo, Buffalo, NY 14214, USA; rideout@buffalo.edu; 3Department of Epidemiology and Environmental Health, School of Public Health and Health Professions, State University of New York at Buffalo, Buffalo, NY 14214, USA; aemillen@buffalo.edu

**Keywords:** dietary pulses, beans, chili, diet, nutrients, dietary intake, early childhood

## Abstract

Bean consumption during childhood may play a role in promoting early-life health given their high nutritional quality. To examine the associations of children’s bean consumption with the socio-demographic characteristics of the child and mother and the child’s nutrient intake, we analyzed data from the WIC-ITFPS-2, which followed children and their mothers at 1, 3, 5, 7, 9, 11, 13, 15, 18, and 24 months (m) following birth. Caregivers (mostly mothers) responded to an interview-administered 24 h recall on their child’s dietary intake at each time point. The intake of dried beans, chili, yellow beans, and lima beans was quantified. Correlate measures included socio-demographic characteristics. Outcome measures of interest focused on the intake of macronutrients (grams and % kcals) and micronutrients at 11 (infancy) and 24 m (toddler) only. To ensure statistical power, we only examined the associations of dried beans and chili with socio-demographics (Chi-square tests) and nutritional outcomes (ANOVA) at 11 and 24 m. The proportion of children who consumed dried beans or chili was very low in the first 6 m of age, started to increase at 7 m (1.2% and 0.4%) and 11 m (4.9% and 2.3%), and reached a high level at 18 m (10.5%) and 24 m (5.9%), respectively. Consumption of yellow or lima beans was rare (<0.1%). At 11 and 24 m, dried bean consumption was higher in children who were White (vs. Black). Dried bean and chili consumption was higher in children who were of Hispanic or Latino ethnicity (vs. non-Hispanic or non-Latino ethnicity). Children who consumed dried beans and chili at 11 or 24 m had a higher intake of total energy, protein, total fiber, potassium, folate, and magnesium compared with non-consumers. The bean consumption was low amongst children, differed by race and ethnicity, and was associated with improved macro- and micronutrient intake in children at 11 and 24 m.

## 1. Introduction

Optimal nutrition throughout infancy and the first 2 years of life has a critical impact on early childhood development and may protect against the programming of metabolic disease risk, triggered by early exposure to both nutritional deficiencies and overnutrition [1]. A previous cross-sectional analysis of data from NHANES 2009–2014 reported that a significant proportion of US children from infancy through preschool age (<5 years) exceeded the recommended limits for sugars, saturated fat, and sodium [2]. Moreover, increasing evidence suggests that an excessive intake of sodium and added sugar during infancy and early childhood is associated with poor cardiovascular health and overweight/obesity status [3,4,5]. This is particularly concerning in light of the wide availability of highly processed foods for infants and children [6,7,8]. Previous work has reported that consumption of added sugars from infant formulas was associated with rapid weight gain in infants and toddlers [9]. Thus, the introduction of whole, nutrient-dense foods during complementary feeding practices and early childhood is recommended to provide optimal nutrition to support the high nutrient intake demands for rapid growth [10], set future flavor preferences and dietary habits [11,12], and improve the nutrition status and health status of children [13].

Pulses such as dried beans, peas, and lentils have an outstanding nutritional profile, containing high-quality protein, complex carbohydrates, micronutrients (including iron, folate, potassium, zinc, and magnesium), and phytochemicals (i.e., catechins and procyanidins) [14,15]. Previous studies in adult populations suggest that pulse consumption is associated with an overall improvement in diet quality due to higher energy-adjusted intakes of fiber, folate, magnesium, potassium, zinc, iron, and choline and a lower intake of fat [16,17,18]. Surprisingly, only limited research has been conducted on early childhood (3–5 years) to assess the health-promoting and disease-prevention benefits of pulse consumption. Further, we are not aware of previous reports that have assessed the prevalence of pulse consumption in infants and children, although it is expected to be low, as pulse consumption amongst the general public is far below recommendations [19,20]. Additionally, the nutrient profiles of bean-based foods may vary significantly depending on factors such as the type of beans that are consumed and cooking methods, as seen in dishes like chili, where various combinations of vegetables and meats can be used to cook them [21]. Little attention has been given to the confounding effects of other nutrients or ingredients that are commonly paired with beans, which may also impact the interpretation of their health effects. Therefore, using data from the Women, Infants, and Children Infant and Toddler Feeding Practices Study-2 (WIC ITFPS-2), we aimed to estimate the prevalence and correlates of bean consumption among children from 1 m to 24 m and describe associations between children’s bean consumption, energy, and nutrient intake at 11 m and 24 m.

## 2. Materials and Methods

### 2.1. Study Population and Sample

This was a secondary data analysis using data from the WIC ITFPS-2. This study focused on infant feeding practices and nutrition outcomes of participants enrolled in the Special Supplemental Nutrition Program for Women, Infants, and Children (WIC). Methodological details on WIC ITFPS-2 have been published previously [22]. Briefly, mothers and their children were followed at 1, 3, 5, 7, 9, 11, 13, 15, 18, 24, 30, 42, 48, 54, and 60 months (m) after birth. Participants (mother-child dyads) were recruited in-person over a 12-week period from 1 July 2013 to 18 November 2013 in the 80 sampled WIC sites (98 mothers per site). The inclusion criteria involved participants who were able to speak English or Spanish (mother), were at least 16 y of age (mother), enrolled in WIC for the first time during the mother’s current pregnancy or before the child was 2.5 m old, had a household income at or below 185 percent of the Federal Poverty Level (the income threshold for WIC participation), and completed an interview when the child was either 1 or 3 m of age. Conversely, the exclusion criteria included infants who were older than 2.5 months at recruitment, mothers under the age of 16, mothers in foster care upon enrollment, and foster parents registering a foster infant. A total of 4489 eligible participants and 4367 children were enrolled in the study, 2322 completed follow-ups at 11 m, and 2461 completed follow-ups at 24 m (Appendix A).

We included 3039 children (Table 1) with complete data on infant consumption of at least one of the 3 common bean products (dried bean, chili, and yellow bean) at 11 or 24 m (two important milestone ages of child development). The mothers provided written informed consent for their and their children’s participation. All study procedures were conducted in accordance with the ethical standards of the responsible committee on social and behavioral science research and with the Helsinki Declaration of 1975, as revised in 2000. This secondary data analysis project used de-identified data from the WIC ITFPS-2. It was approved by the University at Buffalo Institutional Review Board.

### 2.2. Exposures

We examined infant/toddler consumption of several common bean products consumed by the U.S. general population [17], including dried beans, chili, yellow beans, and lima beans using the 24 h data. For each age (1, 3, 5, 9, 11, 13, 15, 18, and 24 m), from the mother-reported list of foods that were consumed by the infant or toddler, children were coded as dried bean consumers if their mother reported that they consumed “dried beans and peas, vegetarian meat substitutes” on the previous day and as non-consumers if the mother reported that they did not consume this food in the previous 24 h. Similarly, three other binary variables for chili, yellow bean, and lima bean consumers/non-consumers were created with the “yes” option if chili (“bean and rice, chili, and other bean mixtures”), yellow bean (“yellow beans”), and lima bean (“immature lima bean”) were consumed, respectively, and with the “no” option if the child did not consume these. Information concerning the frequency or amount of bean intake by infants was not available.

### 2.3. Nutrition Outcome Measures

To estimate an infant’s energy and nutrient intake, a 24 h dietary recall [23,24] interview was conducted with a main caregiver (mostly the mother) over the telephone at each visit using the USDA’s Automated Multi-Pass Method (AMPM) [22,25]. The mother was asked to recall the infant’s dietary intake for each eating event, including foods, beverages, and dietary supplements, from 12:00 a.m. through 11:59 p.m. on the previous day, which could be a weekday or weekend. The caregivers were provided with measuring guides to help them report the child’s portion size during the interview. The USDA Food and Nutrition Database for Dietary Studies (FNDDS), 5.0, was used to assess total calorie and nutrient intake from the 24 h data [25]. If the caregiver did not know about the amount of food that the child ate, then the amount was estimated using the FNDDS [25]. For our nutritional outcome measures, we focused on the nutrients that are known to be rich in bean products (i.e., protein, protein intake (% of energy), carbohydrate, carbohydrate intake (% of energy), total fat, fat intake (% of energy), fiber, iron, potassium, folate, vitamin D, and magnesium) and those that would potentially be displaced from the diet (i.e., saturated fat) with increased dietary bean intake.

### 2.4. Covariates

Based on the literature in this field [17,26,27], we considered the socio-demographic characteristics of children and mothers as the potential correlates of bean consumption at 11 and 24 m in this study. At 1 or 3 m of child age, mothers completed a questionnaire to obtain socio-demographic information. Child characteristics included sex (male, female), race (Black or African American, White, others), ethnicity (Hispanic or Latino, not Hispanic or Latino), and birth weight (low ≤5 lbs. 9 oz.; normal, from 5 lbs. 10 oz. to 9 lbs. 13 oz.; high ≥9 lbs. 14 oz.). Parental characteristics included marital status (married, not married), timing of WIC enrollment [1st trimester, 2nd trimester, 3rd trimester, 4th trimester (postnatal)], body mass index (BMI) status (normal or underweight, overweight, obese), age (16–19 years, 20–25 years, 26 years or older), highest education (9th grade or less, 10th or 11th grade, 12th grade, more than 12th grade), nativity status (born in the U.S., not born in the U.S.), food security score (high or marginal, low, and very low), poverty status (75% of poverty guideline or below, above 75% but no more than 130%, above 130%), cohabitation status (living with father of the baby, not living with father of the baby), maternal breastfeeding duration (no breastfeeding or 0 m, greater than 0 m but ≤3 m, greater than 3 m but ≤6 m, and greater than 6 m), nutrition education (received training on formula only, received training on cereal only, received training on both, received training on neither, not applicable), participation in non-WIC benefit program (not in any other benefit programs, in Supplemental Nutrition Assistance Program [SNAP] or in SNAP and other programs, and in other programs excluding SNAP) [28,29].

### 2.5. Statistical Analysis

To examine the age trend of children’s bean consumption, we plotted bar charts, with the X-axis as the child’s age in months (m) and the Y-axis as the percentage of infants who consumed a specific bean food, i.e., dried beans (Figure 1a), chili (Figure 1b), and yellow beans (Figure 1c).

The percentage of infants who consumed dried beans, chili, and yellow beans by age is presented in Figure 1a–c, respectively.

We examined the frequencies and percentages of our sample characteristics of children and mothers (Table 1).

Next, we classified the children based on their mother-reported bean consumption at 11 and 24 m of child age: children who consumed beans vs. children who did not consume beans. All statistical analyses comparing dried bean consumption to participant characteristics and nutrient intake of the children were conducted based on the dietary intake of the children at 11 and 24 m. We did not examine these associations for the consumption of yellow or lima beans due to insufficient statistical power related to its very low consumption. Chi-square tests were used to examine if the consumption of dried beans and chili at 11 and 24 m (Table 2 and Table 3) differed by socio-demographic characteristics. We conducted ANOVA (analysis of variance) to determine if mean nutrient intakes differed by dried bean consumption (Table 4) and by chili consumption (Table 5). We conducted all analyses using SAS version 9.4 (SAS Institute, Inc., Cary, NC, USA). A two-sided *p*-value < 0.05 was considered statistically significant.

## 3. Results

### 3.1. Sample Characteristics

The distributions of the sample characteristics of children and their mothers are shown in Table 1. The percentages of males (51.0%) and females (49.0%) were similarly distributed in the sample. The children were mostly White (55.8%), non-Hispanic or Latino (58.6%), and had a normal birth weight (91.4%). The majority of the mothers were unmarried (68.7%), were enrolled into WIC in the 1st or the 2nd trimester (71.6%), were overweight or obese (56.4%), were aged 26 years or older (48.0%), had an education level of the 12th grade or lower (61.1%), were born in the U.S. (74.5%), had high or marginal food security (51.2%), were living with the baby’s father (54.6%), breastfed their children (greater than 0 m but ≤3 m [47.9%], greater than 3 m but ≤6 m [11.6%], greater than 6 m [22.6%]), and were participating in SNAP and/or another benefit programs (84.4%).

### 3.2. Age Trend of Bean Consumption

The percentage of infants who consumed dried beans was very low in the first 6 m of age, started to increase at 7 m (1.2%), and peaked at 18 m (10.5%) (Figure 1a). Similarly, the percentage of infants who consumed chili was very low in the first 6 m of age, started to increase at 7 m (0.4%), and peaked at 24 m (5.9%) (Figure 1b). However, the percentage of infants who consumed yellow beans remained very low throughout all ages, with the highest value being only 0.1% at 18 m (Figure 1c).

### 3.3. Socio-Demographics for Infant Dried Bean Consumption

*Dried bean consumption at 11 months.* White children (6.3%) were more likely to consume dried beans than Black children (1.3%) at 11 m of age. Hispanic or Latino children (8.5%) were more likely to consume dried beans than non-Hispanic or Latino children (2.1%) (Table 2). A higher dried bean consumption was observed in children of mothers who were married (6.8% vs. 4.1% children of unmarried mothers), were not born in the U.S. (8.0% vs. 3.8% for those born in the U.S.), and were cohabitating with the infant’s father (5.7% vs. 3.9% of those not cohabitating), or who breastfed their children for a longer duration (0 m [1.9%], greater than 0 m but ≤3 m [4.4%], greater than 3 m but ≤6 m [5.9%], greater than 6 m [8.0%]). Other characteristics, including the child’s sex and birth weight, were not associated with dried bean consumption at 11 m.

*Dried bean consumption at 24 months.* White children (11.8%) were more likely to consume dried beans than Black children (5.2%) at 24 m of age. Hispanic or Latino children (18.7%) were more likely to consume dried beans than non-Hispanic or Latino children (4.1%) (Table 2). Children of married mothers (12.9% vs. 8.6% of children of unmarried mothers) or with maternal education up to the 9th grade (18.5% vs. 8.7% with maternal education above the 12th grade) had a higher intake of dried beans. Children of mothers who were not born in the U.S. (20.7% vs. 6.3% born in the U.S), were not cohabitating with the infant’s father (8.5% vs. 11.2% for cohabitating), breastfed their children for a longer duration (0 m [5.7%], greater than 0 m but ≤3 m [8.8%], greater than 3 m but ≤6 m [11.5%], greater than 6 m [13.7%]), or had participated in SNAP and other programs (8.0% vs. 12.9% participated in programs other than SNAP) had a higher dried bean consumption. Other characteristics, including the child’s sex and birth weight status, were not associated with dried bean consumption at 24 m.

### 3.4. Socio-Demographics for Infant Chili Consumption

*Chili consumption at 11 months.* Hispanic or Latino children (3.8%) were more likely to consume chili at 11 m than non-Hispanic or Latino children (1.1%). Children of mothers with an education higher than the 12th grade (3.3% vs. 1.3% with maternal education of the 12th grade) and children of mothers who were not born in the U.S. (3.5% vs. 1.8% born in the U.S.) had a higher chili consumption (Table 3). Other characteristics, including the child’s sex, race, and birth weight, were not associated with chili consumption at 11 m.

*Chili consumption at 24 months.* Hispanic or Latino children (8.7%) were more likely to consume chili at 24 m than non-Hispanic or Latino children (4.1%). Children of mothers with an education of the 9th grade or less (8.4% vs. 4.5% with maternal education of the 12th grade) and children of mothers who were not born in the U.S. (8.7% vs. 4.9% born in the U.S.) had a higher chili consumption (Table 3). Other characteristics, including the child’s sex, race, and birth weight, were not associated with chili consumption at 24 m.

### 3.5. Nutrient Intake by Child’s Dried Bean Consumption

*Dried bean consumption at 11 months.* Compared with non-consuming children, child consumers of dried beans had a higher total daily energy intake at 11 m (975.05 vs. 907.01 Kcal) (Table 4). A higher protein consumption was observed in dried bean consumers vs. non-consumers (28.79 vs. 24.04 g and 11.4 vs. 10.3% of energy). Although the absolute intake of carbohydrates did not differ, the percentage of energy from carbohydrates was lower in dried bean consumers vs. non-consumers (52.6 vs. 54.1%). In addition, the total fiber intake was higher in consumers vs. non-consumers (8.34 vs. 6.05 g). Dried bean consumption was associated with a higher absolute total fat intake (39.61 vs. 36.51 g), but no difference was observed when this was expressed as a percentage of energy intake. Compared with non-consumers, consumers of dried beans had a higher intake of potassium (1357.65 vs. 1251.84 mg), folate (202.49 vs. 154.39 mcg), and magnesium (137.62 vs. 121.07 mg) but a lower intake of iron (15.38 vs. 18.00 mg) and vitamin D (6.76 vs. 8.02 mcg).

*Dried bean consumption at 24 months.* At 24 m, consumption of dried beans was associated with a lower energy intake (1264.35 vs. 1326.18 Kcal) vs. non-consumption. Compared with non-consumption, dried bean intake was associated with a higher intake of protein (16.7% vs. 15.8% of energy) but a lower intake of fat (42.87 vs. 47.77 g and 30.5% vs. 32.3% of energy). Dried bean consumption was not associated with a difference in total carbohydrate intake; however, a higher intake of total dietary fiber was observed for dried bean consumers (11.56 g) vs. non-consumers (10.38 g). Consumption of dried beans was associated with a higher intake of potassium (2036.85 vs. 1967.00 mg), folate (323.62 vs. 307.21 mcg), and magnesium (195.93 vs. 189.16 mg) compared with non-consumption.

### 3.6. Nutritional Outcomes for Infant Chili Consumption

*Chili consumption at 11 months.* Compared with non-consumption, the consumption of chili was associated with a higher total energy intake at 11 m (1053.67 vs. 907.01 Kcal) (Table 5). A higher protein consumption was noted in chili consumers vs. non-consumers (30.72 vs. 24.12 g and 11.3% vs. 10.4% of energy). Although the percentage of energy from carbohydrates did not differ, the absolute intake of carbohydrates was higher in chili consumers vs. non-consumers (141.77 vs. 122.34 g). In addition, the total fiber intake was higher in consumers vs. non-consumers (9.03 vs. 6.10 g). Chili consumption was associated with a higher absolute total fat intake (41.40 vs. 36.55 g), but no difference was observed when this was expressed as a percentage of energy intake. Compared with non-consumption, chili consumption was associated with higher intakes of potassium (1468.23 vs. 1252.10 mg), folate (206.89 vs. 155.59 mcg), and magnesium (149.47 vs. 121.24 mg). However, the intake of iron and vitamin D did not differ.

*Chili consumption at 24 months.* At 24 m, the consumption of chili was associated with a higher intake of fiber (12.01 vs. 10.40 g). Compared with non-consumption, chili intake was associated with a higher percent of energy from carbohydrates (54.5% vs. 53.5% of energy) but a lower percent of energy from fats (30.9% vs. 32.2% of energy). Chili consumption was not associated with changes in total protein, total carbohydrate, and absolute total fat intake. The consumption of chili was associated with a higher intake of potassium (2085.47 vs. 1967.01 mg) and magnesium (203.39 vs. 188.99 mg) compared with non-consumption.

## 4. Discussion

In this study, we used data from the WIC ITFPS-2 to characterize the prevalence of bean intake among children. We also examined socio-demographic correlates for both mothers and children and the nutrient intake by children’s bean consumption. We observed that the consumption of dried beans and chili in children was low in the first 6 m of age but increased at 7 m and peaked at 18 m and 24 m, respectively. However, the consumption of yellow or lima beans was rare. White and Hispanic or Latino children were found to consume more dried beans at 11 and 24 m than Black and non-Hispanic or Latino ethnicity children. Similarly, the consumption of chili was higher among children who were Hispanic or Latino at 11 and 24 m. Children who consumed dried beans at 11 and 24 m had a higher intake of protein, total dietary fiber, potassium, and folate compared with non-consumers. Children who consumed chili at 11 m had a higher intake of total energy, protein, carbohydrates, total fiber, fat, potassium, folate, and magnesium compared with non-consumers. Similarly, children who consumed chili at 24 m had a higher intake of dietary fiber, and a higher percentage of energy from carbohydrates, potassium, and magnesium.

Dietary recommendations from both the Dietary Guidelines for Americans [30] and the American Heart Association [31] encourage 1.5 cups of cooked pulses/week for both toddlers and children; however, few studies have evaluated childhood pulse consumption patterns in the US [32]. This is unfortunate, as a previous study suggested that although the acceptability of legumes by youths (7–16 years old) varied in general by legume categories, the acceptability of beans and chili with beans was high, at 67% and 66%, respectively [33]. Further, exposing children to nutritious foods during infancy can influence their liking and willingness to consume healthy foods in the longer term [34,35,36]. Our results suggest that the majority of infants and children do not consume bean products in sufficient quantities, with low consumption of dried beans and chili observed across age groups.

This pattern of low bean consumption is likely multifactorial, including issues specifically pertaining to bean foods themselves (i.e., awareness, attitude, and perception issues) and contributing socio-demographic influences (i.e., race and ethnicity, level of education). Previous work on low-income women suggests that mothers typically make food choices based on their availability of time, culture, convenience, and family taste preferences rather than the nutritional composition of food [37]. A hesitancy to include bean foods as a component of complementary feeding practices or as a childhood dietary staple may be related to the general perception that they are hard to digest and are associated with gastrointestinal discomfort [38]. Further, it was previously reported that messaging strategies to position bean products as a low-cost alternative to meat may actually reduce bean consumption amongst low-income women due to their avoidance of ‘cheap’ family foods and the belief that children require animal-sourced proteins [39].

Additionally, we observed several socio-demographic characteristics that influenced the consumption of dried beans and chili in children. Children of mothers who were born in the U.S. had lower dried bean and chili consumption compared with children of mothers who were not born in the U.S. at 11 and 24 m. Further, at 11 and 24 m, dried bean consumption was higher in children who were White (vs. Black) and Hispanic or Latino ethnicity (vs. non-Hispanic or Latino ethnicity). We observed that children who were breastfed for a duration of more than 6 m had a higher dried bean consumption at 11 m and 24 m, respectively. A recent analysis of population trends in pulse consumption using data from the 2003–2014 National Health and Nutrition Examination Survey reported that Hispanic and Mexican American adults (≥19 years) were more likely to consume dietary pulses compared with other ethnic groups [16]. This may be related to ethnic differences in cooking practices, as Winham et al., (2019) reported that compared with low-income Latina women, non-Hispanic White women may lack a general awareness of how to properly cook beans [26]. Interestingly, we also observed that children (24 m) of mothers who had an education up to the 9th grade or less had a higher intake of dried beans compared with those who had an education of the 12th grade. This finding was perhaps surprising, as previous work has reported that adults with an education beyond high school were more likely to consume pulses compared to those with less education [16,40]. Although the reason(s) for this discrepancy is not clear, it does not appear to be related to maternal poverty status, a finding that is supported by previous work reporting that pulse consumption was not associated with income level [40,41]. These findings suggest that socio-demographic factors, including children’s race and ethnicity, mother’s level of education, nativity status, and breastfeeding duration, play a critical role in determining pulse consumption patterns in young children.

As a low-calorie and nutrient-dense food, pulses are an excellent nutrient source to support childhood development and health. Based on the nutrient composition, pulses can contribute significantly to childhood requirements for protein, dietary fiber, and micronutrients including iron, zinc, and magnesium [32]. In addition to their contribution of nutrients, the consumption of dietary pulses may help to improve the overall diet quality by displacing less healthy nutrients of concern (i.e., added sugar and saturated fat). Accordingly, we observed that compared with non-consumers, children who consumed dried beans at 11m had a higher intake of total energy, protein, total fiber, potassium, folate, and magnesium. Alternatively, the % of energy from carbohydrates was lower in consumers vs. non-consumers at 11 m. The findings at 24 m were similar for micronutrient intake including potassium, folate, and magnesium, but differed with respect to energy (1264 vs. 1326 kcal) and fat (30.5 vs. 32.3% energy), which were both lower in consumers versus non-consumers. Associations in nutritional outcomes for chili consumption at 11 and 24 m were largely discordant, with similarities only noted for fiber and magnesium, which were both higher in consumers vs. non-consumers. Compared with non-consumers, chili consumption at 11 m was associated with an increased intake of energy and protein (% energy); however, these associations were not evident at 24 m. However, chili consumption at 24 m was associated with an increased carbohydrate intake (% energy) and a reduced fat intake (% energy).

We are not aware of any previous work that has examined the association of pulse intake with diet quality or nutrient intake parameters in U.S. children for direct comparison. However, a previous study of 772 mother-child pairs in Southern Ethiopia examined the influence of a maternal nutrition education program that was meant to increase education about the benefits of using pulses in complementary feeding practices [42]. The intervention resulted in a substantial increase in maternal knowledge, attitude, and practices towards pulses and increased the frequency of pulse intake and diet diversity scores in children. Further, anthropometric indices, including stunting, wasting, and underweight were all improved in the intervention group compared with the control. Beyond this study, the influence of increased pulse consumption on childhood health outcomes is not known. However, in previous adult studies, the consumption of pulses and pulse-enriched products has been shown to reduce blood pressure [43], improve glycemic control [44], increase satiety [45], and help in the maintenance of a healthy body weight [46,47]. Further, a recent meta-analysis assessing the relationship between dietary pulses and cardiometabolic health reported a reduced incidence of cardiovascular disease and coronary heart disease with pulse consumption but with low certainty, necessitating the need for future research [48].

## 5. Strengths and Limitations

The strengths of this study included the prospective design and relatively long follow-up. However, secondary data analysis is constrained by various limitations, including limited control over the data collection, data quality issues, ethical considerations, and restrictions in addressing new research questions. Our study also had several other limitations. Firstly, the primary limitation might be social desirability bias or error from a lack of memory recalling diet, as the mothers reported the children’s diet through a 24 h dietary recall interview. Secondly, we might have limited statistical power to detect significant associations due to the moderate sample size of children who consumed beans. Thirdly, we lacked information about the frequency or amount of bean intake by infants. Additionally, we did not assess the average grams/cups of bean foods, which may have provided valuable insight into the dietary habits of the participants. Fourthly, the 24 h for an infant’s dietary intake could be a weekday or weekend, which may influence what the child is eating. Fifthly, people may cook chili differently, with different types of vegetables and meat. However, we did not have any information on this variation and how different types of beans, vegetables, or meats in the chili may impact infant nutritional outcomes. Furthermore, we did not calculate the percentage of children that met or did not meet the Dietary Reference Intakes by pulse consumption status to determine the number of infants who were below recommended intakes of macro- and micronutrients. However, these aspects would be considered in future analyses.

## 6. Conclusions

In summary, the results of this study suggest that the consumption of bean products, particularly dried beans, and chili, was low during childhood, differed by race and ethnicity in children at 11 m and 24 m, and was associated with improved macro- and micronutrient intake in early childhood. Future work should examine how bean consumption during infancy and early childhood influences longer-term food preferences and health outcomes.

## Figures and Tables

**Figure 1 nutrients-16-01120-f001:**
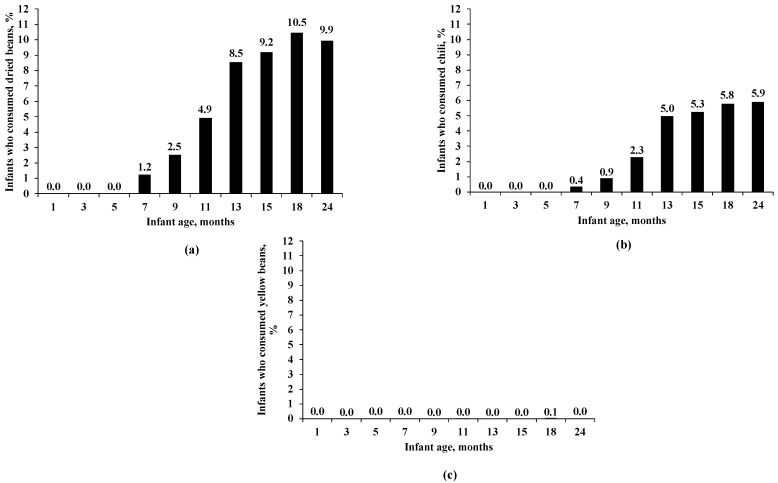
(**a**) Percentage of infants who consumed dried beans by age. (**b**) Percentage of infants who consumed chili by age. (**c**) Percentage of infants who consumed yellow beans by age. The available sample size varied by infant age for dried beans, chili, and yellow beans: 3398 for 1 m, 2881 for 3 m, 2631 for 5 m, 3121 for 7 m, 2443 for 9 m, 2316 for 11 m, 2799 for 13 m, 2076 for 15 m, 1998 for 18 m, and 2465 for 24 m.

**Table 1 nutrients-16-01120-t001:** Distribution of characteristics of children and mothers in the analytic sample.

	N = (3039)
Sample Characteristics *	*n* (%)
**Child’s sex**	
Male	1549 (51.0)
Female	1490 (49.0)
**Child’s race**	
Black or African American	814 (27.3)
White	1664 (55.8)
All other	505 (16.9)
**Child’s ethnicity**	
Hispanic or Latino	1245 (41.4)
Not Hispanic or Latino	1763 (58.6)
**Child’s birth weight**	
Low	223 (7.3)
Normal	2778 (91.4)
High	38 (1.3)
**Maternal marital status**	
Married	950 (31.3)
Not married	2089 (68.7)
**Maternal timing of WIC enrollment**	
1st trimester	969 (31.9)
2nd trimester	1205 (39.7)
3rd trimester	461 (15.2)
Postnatal	404 (13.3)
**Maternal BMI at screening**	
Normal or underweight	1327 (43.7)
Overweight	822 (27.1)
Obese	890 (29.3)
**Maternal age of enrollment**	
16–19 years	336 (11.1)
20–25 years	1243 (40.9)
26 years or older	1460 (48.0)
**Maternal highest education**	
9th grade or less	282 (9.3)
10th or 11th grade	431 (14.2)
12th grade	1140 (37.6)
More than 12th grade	1179 (38.9)
**Maternal nativity status**	
Mother born in U.S.	2262 (74.5)
Mother not born in U.S.	776 (25.5)
**Maternal food security score**	
High or marginal food security	1555 (51.2)
Low food security	956 (31.5)
Very low food security	528 (17.4)
**Maternal poverty status**	
75% of poverty guideline or below	1914 (63.0)
Above 75% but no more than 130% of poverty guideline	826 (27.2)
Above 130% of poverty guideline	299 (9.8)
**Maternal cohabitation status**	
Mother living with father of the baby	1657 (54.6)
Mother not living with father of the baby	1379 (45.4)
**Maternal breastfeeding duration**	
No breastfeeding or 0 m	526 (17.9)
Greater than 0 m but ≤3 m	1407 (47.9)
Greater than 3 m but ≤6 m	340 (11.6)
Greater than >6 m	665 (22.6)
**Maternal nutrition education**	
Received training on formula only	552 (18.2)
Received training on cereal only	289 (9.5)
Received training on both	1574 (51.8)
Received training on neither	549 (18.1)
Not applicable	75 (2.5)
**Maternal benefit program non-WIC**	
Does not participate in any other benefit programs	476 (15.7)
Participates in SNAP or in SNAP and other programs	1518 (50.0)
Participates in other programs excluding SNAP	1045 (34.4)

WIC, Special Supplemental Nutrition Program for Women, Infants, and Children; BMI, body mass index; SNAP, Supplemental Nutrition Assistance Program. * The sum of categories might be smaller than the total due to missing data.

**Table 2 nutrients-16-01120-t002:** Associations between socio-demographics and dried bean consumption at 11 and 24 m.

	Dried Beans Consumed at 11 m	*p*-Value	Dried Beans Consumed at 24 m	*p*-Value
	*n* (%)		*n* (%)	
**Child’s sex**				
Male	58 (4.9)	0.998	136 (10.9)	0.123
Female	56 (4.9)		109 (9.0)	
**Child’s race**				
Black or African American	7 (1.3)	<0.001	37 (5.2)	<0.001
White	84 (6.3)		154 (11.8)	
All other	20 (5.1)		47 (11.6)	
**Child’s ethnicity**				
Hispanic or Latino	86 (8.5)	<0.001	183 (18.7)	<0.001
Not Hispanic or Latino	27 (2.1)		60 (4.1)	
**Child’s birth weight**				
Low	8 (4.9)	0.454	12 (6.7)	0.265
Normal	106 (5.0)		229 (10.2)	
High	0 (0.0)		4 (13.3)	
**Maternal marital status**				
Married	50 (6.8)	0.005	100 (12.9)	0.001
Not married	64 (4.1)		145 (8.6)	
**Maternal timing of WIC enrollment**				
1st trimester	43 (5.5)	0.635	95 (11.9)	0.072
2nd trimester	43 (4.6)		96 (9.9)	
3rd trimester	18 (5.3)		29 (7.9)	
Postnatal	10 (3.7)		25 (7.7)	
**Maternal BMI at screening**				
Normal or underweight	47 (4.5)	0.257	109 (10.3)	0.866
Overweight	35 (6.2)		66 (9.7)	
Obese	32 (4.6)		70 (9.6)	
**Maternal age of enrollment**				
16–19 years	12 (4.7)	0.970	27 (10.5)	0.174
20–25 years	46 (4.9)		84 (8.6)	
26 years or older	56 (5.0)		134 (10.9)	
**Maternal highest education**				
9th grade or less	17 (7.5)	0.240	42 (18.5)	<0.001
10th or 11th grade	18 (5.4)		36 (10.7)	
12th grade	37 (4.3)		80 (8.8)	
More than 12th grade	42 (4.7)		86 (8.7)	
**Maternal nativity status**				
Mother born in U.S.	64 (3.8)	<0.001	115 (6.3)	<0.001
Mother not born in U.S.	50 (8.0)		130 (20.7)	
**Maternal food security score**				
High or marginal food security	59 (4.9)	0.109	130 (10.5)	0.194
Low food security	43 (5.9)		82 (10.4)	
Very low food security	12 (3.1)		33 (7.6)	
**Maternal poverty status**				
75% of poverty guideline or below	70 (4.8)	0.737	154 (9.9)	0.521
Above 75% but no more than 130%	30 (4.7)		72 (10.7)	
Above 130% of poverty guideline	14 (6.0)		19 (8.1)	
**Maternal cohabitation status**				
Mother living with father of the baby	74 (5.7)	0.046	149 (11.2)	0.024
Mother not living with father of the baby	40 (3.9)		96 (8.5)	
**Maternal breastfeeding duration**				
No breastfeeding or 0 m	7 (1.9)	<0.001	24 (5.7)	<0.001
Greater than 0 m but ≤3 m	48 (4.4)		98 (8.8)	
Greater than 3 m but ≤6 m	15 (5.9)		33 (11.5)	
Greater than 6 m	42 (8.0)		78 (13.7)	
**Maternal nutrition education**				
Received training on formula only	25 (5.8)	0.718	50 (11.2)	0.595
Received training on cereal only	12 (5.3)		29 (11.8)	
Received training on both	51 (4.3)		119 (9.4)	
Received training on neither	23 (5.6)		43 (9.4)	
Not applicable	3 (4.5)		4 (7.0)	
**Maternal benefit program non-WIC**				
Did not participate in any programs	21 (5.5)	0.691	36 (9.8)	0.001
Participates in SNAP and other programs	52 (4.5)		98 (8.0)	
Participates in programs other than SNAP	41 (5.2)		111 (12.9)	

WIC, Special Supplemental Nutrition Program for Women, Infants, and Children; BMI, body mass index; SNAP, Supplemental Nutrition Assistance Program.

**Table 3 nutrients-16-01120-t003:** Associations between socio-demographics and chili consumption at 11 m and 24 m.

	Chili Consumed at 11 m	*p*-Value	Chili Consumed at 24 m	*p*-Value
	*n* (%)		*n* (%)	
**Child’s sex**				
Male	22 (1.9)	0.168	67 (5.4)	0.218
Female	31 (2.7)		79 (6.5)	
**Child’s race**				
Black or African American	14 (2.5)	0.851	42 (5.9)	0.538
White	29 (2.2)		71 (5.4)	
All other	8 (2.0)		28 (6.9)	
**Child’s ethnicity**				
Hispanic or Latino	38 (3.8)	<0.001	85 (8.7)	<0.001
Not Hispanic or Latino	14 (1.1)		60 (4.1)	
**Child’s birth weight**				
Low	3 (1.8)	0.859	12 (6.7)	0.763
Normal	49 (2.3)		133 (5.9)	
High	1 (3.3)		1 (3.3)	
**Maternal marital status**				
Married	15 (2.0)	0.569	37 (4.8)	0.106
Not married	38 (2.4)		109 (6.4)	
**Maternal timing of WIC enrollment**				
1st trimester	18 (2.3)	0.593	47 (5.9)	0.798
2nd trimester	18 (1.9)		54 (5.5)	
3rd trimester	11 (3.2)		22 (6.0)	
Postnatal	6 (2.2)		23 (7.1)	
**Maternal BMI at screening**				
Normal or underweight	25 (2.4)	0.953	74 (7.0)	0.133
Overweight	13 (2.3)		33 (4.8)	
Obese	15 (2.2)		39 (5.4)	
**Maternal age of enrollment**				
16–19 years	5 (2.0)	0.886	19 (7.4)	0.562
20–25 years	23 (2.4)		58 (5.9)	
26 years or older	25 (2.2)		69 (5.6)	
**Maternal highest education**				
9th grade or less	7 (3.1)	<0.001	19 (8.4)	<0.001
10th or 11th grade	6 (1.8)		18 (5.4)	
12th Grade	11 (1.3)		41 (4.5)	
More than 12th grade	29 (3.3)		68 (6.9)	
**Maternal nativity status**				
Mother born in U.S.	31 (1.8)	0.017	90 (4.9)	<0.001
Mother not born in U.S.	22 (3.5)		55 (8.7)	
**Maternal food security score**				
High or marginal food security	23 (1.9)	0.273	66 (5.3)	0.425
Low food security	22 (3.0)		53 (6.7)	
Very low food security	8 (2.0)		27 (6.2)	
**Maternal poverty status**				
75% of poverty guideline or below	35 (2.4)	0.296	104 (6.7)	0.098
Above 75% but no more than 130%	16 (2.5)		33 (4.9)	
Above 130% of poverty guideline	2 (0.9)		9 (3.9)	
**Maternal cohabitation status**				
Mother living with father of the baby	32 (2.5)	0.503	72 (5.4)	0.244
Mother not living with father of the baby	21 (2.1)		74 (6.5)	
**Maternal breastfeeding duration**				
No breastfeeding or 0 m	6 (1.6)	0.387	18 (4.2)	0.045
Greater than 0 m but ≤ 3 m	23 (2.1)		66 (5.9)	
Greater than 3 m but ≤ 6 m	9 (3.5)		14 (4.9)	
Greater than 6 m	14 (2.7)		47 (8.3)	
**Maternal nutrition education**				
Received training on formula only	12 (2.8)	0.265	23 (5.2)	0.170
Received training on cereal only	4 (1.8)		22 (9.0)	
Received training on both	25 (2.1)		74 (5.9)	
Received training on neither	8 (1.9)		26 (5.7)	
Not applicable	4 (6.0)		1 (1.8)	
**Maternal benefit program non-WIC**				
Did not participate in any programs	9 (2.3)	0.943	22 (6.0)	0.076
Participates in SNAP and other programs	25 (2.2)		85 (6.9)	
Participates in programs other than SNAP	19 (2.4)		39 (4.5)	

WIC, Special Supplemental Nutrition Program for Women, Infants, and Children; BMI, body mass index; SNAP, Supplemental Nutrition Assistance Program.

**Table 4 nutrients-16-01120-t004:** Association of dried bean consumption with nutritional outcomes in children at 11 and 24 m.

Nutritional Outcomes	Dried Bean at 11 m			Dried Bean at 24 m		
	No Consumption (*n* = 2202)	Consumption (*n* = 114)	*p*-Value	No Consumption (*n* = 2193)	Consumption (*n* = 241)	*p*-Value
	Mean ± SD	Mean ± SD		Mean ± SD	Mean ± SD	
Energy (Kcal)	907.01 ± 330.97	975.05 ± 340.89	0.033	1326.18 ± 326.31	1264.35 ± 316.82	0.005
Protein (g)	24.04 ± 12.93	28.79 ± 15.40	<0.001	51.85 ± 13.42	51.97 ± 12.84	0.896
Protein intake (% of energy)	10.3 ± 3.3	11.4 ± 3.3	0.001	15.8 ± 2.5	16.7 ± 2.5	<0.001
Carbohydrate (g)	122.50 ± 47.64	128.26 ± 47.96	0.209	176.14 ± 43.00	170.85 ± 45.38	0.072
Carbohydrate intake (% of energy)	54.1 ± 8.0	52.6 ± 7.3	0.048	53.5 ± 5.7	54.2 ± 5.7	0.052
Total fat (g)	36.51 ± 14.78	39.61 ± 14.62	0.029	47.77 ± 14.31	42.87 ± 13.06	<0.001
Fat intake (% of energy)	36.4 ± 7.0	36.9 ± 6.6	0.438	32.3 ± 4.7	30.5 ± 5.1	<0.001
Fiber (g)	6.05 ± 4.05	8.34 ± 5.00	<0.001	10.38 ± 3.35	11.56 ± 3.37	<0.001
Iron (mg)	18.00 ± 13.84	15.38 ± 13.18	0.048	11.78 ± 3.31	11.52 ± 3.27	0.255
Potassium (mg)	1251.84 ± 487.74	1357.65 ± 552.37	0.025	1967.00 ± 435.80	2036.85 ± 428.88	0.018
Folate (mcg)	154.39 ± 93.84	202.49 ± 119.73	<0.001	307.21 ± 115.34	323.62 ± 116.28	0.036
Vitamin D (mcg)	8.02 ± 5.26	6.76 ± 4.47	0.012	8.18 ± 3.09	8.44 ± 2.97	0.216
Magnesium (mg)	121.07 ± 60.36	137.62 ± 66.77	0.005	189.16 ± 40.66	195.93 ± 38.68	0.014

**Table 5 nutrients-16-01120-t005:** Association of chili consumption with nutritional outcomes in children at 11 and 24 m.

Nutritional Outcomes	Chili at 11 m			Chili at 24 m		
	No Consumption (*n* = 2263)	Consumption (*n* = 53)	*p*-Value	No Consumption (*n* = 2292)	Consumption (*n* = 142)	*p*-Value
	Mean ± SD	Mean ± SD		Mean ± SD	Mean ± SD	
Energy (Kcal)	907.01 ± 329.87	1053.67 ± 379.44	0.002	1318.46 ± 325.74	1345.89 ± 327.65	0.330
Protein (g)	24.12 ± 12.97	30.72 ± 16.65	<0.001	51.80 ± 13.34	52.77 ± 13.75	0.401
Protein intake (% of energy)	10.4 ± 3.3	11.3 ± 3.0	0.045	15.9 ± 2.5	15.8 ± 2.3	0.691
Carbohydrate (g)	122.34 ± 47.51	141.77 ± 50.31	0.003	175.21 ± 43.27	182.14 ± 42.74	0.064
Carbohydrate intake (% of energy)	54.0 ± 7.9	54.2 ± 7.6	0.891	53.5 ± 5.7	54.5 ± 5.3	0.038
Total fat (g)	36.55 ± 14.69	41.40 ± 18.11	0.018	47.34 ± 14.29	46.44 ± 13.86	0.468
Fat intake (% of energy)	36.5 ± 7.0	35.4 ± 7.1	0.281	32.2 ± 4.8	30.9 ± 4.6	0.003
Fiber (g)	6.10 ± 4.04	9.03 ± 6.38	<0.001	10.40 ± 3.31	12.01 ± 4.00	<0.001
Iron (mg)	17.87 ± 13.82	18.19 ± 14.16	0.868	11.74 ± 3.33	12.01 ± 3.01	0.348
Potassium (mg)	1252.10 ± 489.83	1468.23 ± 521.25	0.002	1967.01 ± 431.80	2085.47 ± 479.91	0.002
Folate (mcg)	155.59 ± 94.33	206.89 ± 137.76	<0.001	307.76 ± 115.53	326.10 ± 114.26	0.066
Vitamin D (mcg)	7.96 ± 5.24	8.04 ± 5.08	0.910	8.21 ± 3.08	8.15 ± 2.99	0.807
Magnesium (mg)	121.24 ± 60.16	149.47 ± 79.13	0.001	188.99 ± 40.00	203.39 ± 46.10	<0.001

## Data Availability

The data presented in this study cannot be shared without permission from the USDA (data owner). Researchers who are interested in using the WIC-ITFPS-2 data can contact the USDA directly.

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
