# Peer review of "Bean Consumption during Childhood Is Associated with Improved Nutritional Outcomes in the First Two Years of Life"

_nutrients, 2024, doi:10.3390/nu16081120_

Round 1

Reviewer 1 Report

Comments and Suggestions for Authors

Brief Summary

This study examined the associations between child bean (dried beans, chili, yellow beans, lima beans) intake and child and mother sociodemographic variables and child nutrient intake using longitudinal 24-h recall data from the WIC-14 ITFPS-2 from birth to 24 months. Results showed that children who consumed dried beans or chili was very low in the first 6 m of age and increased over time. Intakes were higher in those of Hispanic or Latino ethnicity and were related to higher intakes of  energy, protein, total fiber, potassium, folate, and magnesium.

Feedback

Introduction

It is not clear why the second word is capitalized (Optimal)

The introduction provides a solid rationalize for the study

 Methods

The statement “Consent from participants was not needed, since it 94 was determined as non-human research” could be debated. Perhaps the ethics committee deemed the research data anonymous.  It is clearly data from humans.  It is recommended that this statement simply not be included.

 Lines 120-22: it would be helpful to include the complete list of nutrients that were considered

 For the identification of bean consumption, was amount considered?  Or would any amount identify the participants as a consumer?  Please include this detail.

 Some clarification for yellow beans would be helpful.  Are these yellow dried beans?  Or yellow wax beans?  As would yellow wax beans be more like a fresh green bean and not a type of dried bean?

 Results

For Tables 4 and 5, can you please include the sample size for the each of the 2 groups?

Discussion

Lines 360-1: Please take care with the use of “influence” since the current study was not cause and effect

Author Response

Reviewer comment: (Introduction)

It is not clear why the second word is capitalized (Optimal)

Author Response:

Line 35: The capitalization of “optimal” has been corrected.

Reviewer comment: (Methods)
The statement “Consent from participants was not needed, since it 94 was determined as non-human research” could be debated. Perhaps the ethics committee deemed the research data anonymous.  It is clearly data from humans.  It is recommended that this statement simply not be included.
Author Response:

The statement regarding consent for non-human research has been removed.

Reviewer comment:

Lines 120-22: it would be helpful to include the complete list of nutrients that were considered

For the identification of bean consumption, was amount considered?  Or would any amount identify the participants as a consumer?  Please include this detail.

Some clarification for yellow beans would be helpful.  Are these yellow dried beans?  Or yellow wax beans?  As would yellow wax beans be more like a fresh green bean and not a type of dried bean?

Author Response:

Line 125-127: A comprehensive list of considered nutrients has been included.

Line 110-111: In the methods section under exposures, we have added a line to address this concern.

The dataset or codebook does not provide any details on yellow beans.

Reviewer comment: (Results)

For Tables 4 and 5, can you please include the sample size for the each of the 2 groups?

Author Response:

The sample size for each of the two groups (consumers vs non-consumers) has been added to Tables 4 and 5.

Reviewer comment: (Discussion)

Lines 360-1: Please take care with the use of “influence” since the current study was not cause and effect

Author Response:

Line 369-370: The term “influence” has been replaced with “Association”.

Reviewer 2 Report

Comments and Suggestions for Authors

Dear Authors:

Abstract: Please check "Dried bean or chili consumption was higher in children who were of Hispanic or Latino ethnicity (vs non-Hispanic or 26 Latino ethnicity)." Something looks wrong.

Please use uniform and consistent abbreviations (for instance, 24-hour recall, etc.). Also, once abbreviated, it is not needed to use the expanded form (for instance, WIC ITFPS-2)

Introduction section:

Lines 67-69, it is stated that "we aimed to estimate the prevalence and correlates of bean consumption among children from 1 m to 24 m and describe associations between child bean consumption, energy, and nutrient intake at 11 m and 24 m", while within the abstract you mention have studied associations at 1, 3, 5, 7, 9, 11, 13, 15, 18, and 24 months (m).

Results: I appreciate that you have included p. values in Table 2. But I suggest adding p values in Table 1 will be helpful too.

Methodology:

Please clarify (using some references showing that a single 24-hour recall can be considered valid for such a purpose.  

Lines 133-137: please cite references for criteria or operational definitions you have used.

Limitations: line s377-378: It is stated within the paper that "The strengths of this study included the prospective design and relatively long follow-up", however even if the design of the primary study was prospective –which is not mentioned before in the paper- please consider the fact that using secondary analysis has its own limitations and issues. This should be added to the paper and discussed. See for instance "Limitations and recommendations for use of secondary data analysis in pediatric research, https://doi.org/10.1080/02739615.2023.2279064

Author Response

Reviewer comment: (Abstract)

Please check "Dried bean or chili consumption was higher in children who were of Hispanic or Latino ethnicity (vs non-Hispanic or 26 Latino ethnicity)." Something looks wrong.

Please use uniform and consistent abbreviations (for instance, 24-hour recall, etc.). Also, once abbreviated, it is not needed to use the expanded form (for instance, WIC ITFPS-2)

Author Response:

Line 26-27: Dried bean and chili consumption was higher in children who were of Hispanic or Latino ethnicity (vs non-Hispanic or non-Latino ethnicity). The wording has been corrected. Additionally, consistent abbreviations, such as "24-hour recall" (WIC ITFPS-2), have been applied uniformly across the manuscript.

Reviewer comment: (Introduction section)

Lines 67-69, it is stated that "we aimed to estimate the prevalence and correlates of bean consumption among children from 1 m to 24 m and describe associations between child bean consumption, energy, and nutrient intake at 11 m and 24 m", while within the abstract you mention have studied associations at 1, 3, 5, 7, 9, 11, 13, 15, 18, and 24 months (m).

Author response:

 Line 19-20: It has been clarified in the abstract.

Reviewer comment: (Results)

I appreciate that you have included p. values in Table 2. But I suggest adding p values in Table 1 will be helpful too.

Author response:

In Table 1, we do not have any comparison groups, so we cannot add a p-value.

Reviewer comment: (Methodology)

Please clarify (using some references showing that a single 24-hour recall can be considered valid for such a purpose.  

Lines 133-137: please cite references for criteria or operational definitions you have used.

Author response:

I have cited two papers (references: 23,24) to show that a single 24-hour recall can be considered valid for such a purpose. Similarly, I have cited additional two papers (references: 29,30) to clarify the criteria or operational definitions that I have used in the manuscript.

Reviewer comment: (Limitations)

Lines 377-378: It is stated within the paper that "The strengths of this study included the prospective design and relatively long follow-up", however even if the design of the primary study was prospective –which is not mentioned before in the paper- please consider the fact that using secondary analysis has its own limitations and issues. This should be added to the paper and discussed. 

Author response:

Line 387-389: A line has been added to the limitations section, considering the fact that using secondary data analysis has its own limitations and issues.

Reviewer 3 Report

Comments and Suggestions for Authors

Summary:

Abstract: The study investigates the nutritional effects of bean consumption in early childhood associated with the mother's sociodemographic characteristics. The results of the study demonstrated that the consumption of beans and pepper was higher at 11 or 24 months of age, and that they obtained a high value of total energy, protein, total fiber, potassium, folate and magnesium, compared to children who did not consume them. Overall, bean intake was low among children, which differed by race and ethnicity and was associated with improved macro and micronutrient intake between 11 and 24 months of age, thus considering bean consumption in early childhood a multifactorial issue including sociodemographic influences. The manuscript is relevant, but some points can be clarified, as highlighted below.

Comments:

Keywords

Authors must include “Dietary intake” as keywords.

Introduction section

1.     Do you have current statistical data on the incidence of legume consumption among children and adults in the USA? If so, the authors could include it to score prevalence.

Materials and Methods section

1.     The authors must describe the study’s inclusion and exclusion criteria.

Discussion section

1.     The authors must point out and describe the states that were associated with ethnicities, correlating with skin color and their dietary consumption of beans.

Author Response

Reviewer comment: (Keywords)

Authors must include “Dietary intake” as keywords.

Author response:

I have added “Dietary intake” to the keywords as requested.

Reviewer comment: (Introduction section)

  1. Do you have current statistical data on the incidence of legume consumption among children and adults in the USA? If so, the authors could include it to score prevalence.

Author response:

In the study, we examined the exposure to two types of bean products, specifically "dried beans" and "chili," focusing solely on the time points of 11 and 24 months. Graphs illustrating the prevalence of dried beans (Figure 1.a) and chili (Figure 1.b) at these time points are presented in the manuscript.

Reviewer comment: (Materials and Methods section)

  1. The authors must describe the study’s inclusion and exclusion criteria.

Author response:

Line 80-87: The inclusion and exclusion criteria have been added to the methods section.

Reviewer comment: (Discussion section)

  1. The authors must point out and describe the states that were associated with ethnicities, correlating with skin color and their dietary consumption of beans.

Author response:

Unfortunately, we do not have information on this.

Round 2

Reviewer 2 Report

Comments and Suggestions for Authors

My comments are addressed.